# The Effect of Geoclimatic Factors on the Distribution of Paracoccidioidomycosis in Mato Grosso do Sul, Brazil

**DOI:** 10.3390/jof10030165

**Published:** 2024-02-21

**Authors:** Larissa Rodrigues Fabris, Nathan Guilherme de Oliveira, Bruna Eduarda Bortolomai, Lavínia Cássia Ferreira Batista, Marcos Henrique Sobral, Alisson André Ribeiro, Ursulla Vilella Andrade, Antonio Conceição Paranhos Filho, Lídia Raquel de Carvalho, Ida Maria Foschiani Dias Baptista, Rinaldo Poncio Mendes, Anamaria Mello Miranda Paniago

**Affiliations:** 1Infectious and Parasitic Diseases, School of Medicine, Federal University of Mato Grosso do Sul, Campo Grande 79070-900, MS, Brazil; larissafabris@ibest.com.br (L.R.F.); geotec.ribeiro@gmail.com (A.A.R.); ursulla1@gmail.com (U.V.A.); toniparanhos@gmail.com (A.C.P.F.); anapaniago@yahoo.com.br (A.M.M.P.); 2Lauro de Souza Lima Institute, Bauru 17034-971, SP, Brazil; ng.oliveira@unesp.br (N.G.d.O.); bruna.bortolomai@unesp.br (B.E.B.); lavinia.ferreira@unesp.br (L.C.F.B.); marcoshsobral@gmail.com (M.H.S.); 3Tropical Diseases, School of Medicine, São Paulo State University—UNESP, Botucatu 18618-687, SP, Brazil; lidia.carvalho@unesp.br (L.R.d.C.); tietemendes@terra.com.br (R.P.M.)

**Keywords:** paracoccidioidomycosis, *Paracoccidioides* species, environment, Mato Grosso do Sul state, climate, environment and paracoccidioidomycosis

## Abstract

The incidence of paracoccidioidomycosis (PCM) varies in Latin America, and it is influenced by environmental factors. This study evaluated the distribution of PCM acute/subacute form (AF) cases and their correlation with geoclimatic factors in the Mato Grosso do Sul (MS) state. The study included 81 patients diagnosed with the PCM/AF at the University Hospital of the Federal University of Mato Grosso do Sul between January 1980 and February 2022. Geographic coordinates, health microregion of patient’s residence, compensated average temperature, relative air humidity (RH), El Niño Southern Oscillation (ENSO), and average global temperature were analyzed. The highest incidence was observed in the Aquidauana (7/100,000 inhabitants), while Campo Grande, the state’s capital, had the highest number (n = 34; 42.4%) and density (4.4 cases/km^2^) of cases. The number of cases increased during extended periods of the El Niño phenomenon. A positive correlation was found between higher RH and PCM/AF cases. Most PCM/AF cases were found in areas with loamy soils and RH ranging from 60.8 to 73.6%. In MS, the health microregions of PCM/AF patients are characterized by deforestation for agricultural and pasture use, coupled with loamy soils and specific climatic phenomena leading to higher soil humidity.

## 1. Introduction 

Paracoccidioidomycosis (PCM) is a systemic mycotic disease that came to medical attention in 1908 through the publication of two cases by Adolpho Lutz [1]. Lutz reported the cases, described the histopathological lesions (typical tubercle-like nodules), isolated the etiologic agent and differentiated it from *Coccidioides immitis*, reproduced the disease in an experimental model using guinea pigs, and identified the thermal dimorphism. Few researchers covered all these points in reporting a new disease. However, the name of the etiologic agent, *Paracoccidioides brasiliensis*, was proposed only in 1930 by Almeida [2]. Recent studies using molecular methodology identified several clades that resulted in the *P. brasiliensis* complex, with four cryptic species—*P brasiliensis stricto sensu* (S1), broadly distributed through Latin America; *P. Americana* (PS2), reported in Brazil and Venezuela; *P. restrepiensis* (PS3), restricted to clinical cases from Colombia; and *P. venezuelensis* (PS4), recently suggested and recovered from Venezuelan patients. As Pb01 and Pb01-like isolates, originating from the Midwest region of Brazil and one from Ecuador, could not be grouped into any of these species, they were located in a new species that was named *Paracoccidioides lutzii* [3,4,5].

The disease presents two main clinical forms: the chronic form (CF), which has a long duration of symptomatology (≥4–6 months) and affects mostly rural workers over 30 years old, and the acute/subacute form (AF), which has a shorter duration of symptomatology (≤2 months) and involves younger individuals, including children, adolescents, and young adults [6]. The AF is therefore a better indicator of the site of *Paracoccidioides* infection than the CF, since the younger affected individuals usually remain at the same residence.

The infection progress and the different clinical forms are highly determined by the host’s immune response, the virulence of the fungus, as well as environmental biotic and abiotic components [6,7,8].

PCM is exclusively found and widely distributed throughout Latin America, with the majority of cases occurring in Brazil [9]. Studies carried out by Chirife and Del Rio (1965) [10] and Borelli (1972) [11] were the first to evaluate the geographic parameters of the areas where the patients came from—moderately warm and humid, with short and not extremely cold winters. Evaluation of the incidence rate according to different Brazilian regions strongly suggested the influence of the geoclimatic conditions [12]. *P. brasiliensis* is found in the soil of tropical or subtropical regions, which are characterized by an abundance of watercourses, temperatures that range between 10 °C and 28 °C, high annual precipitation (between 500 and 2500 mm), and an acidic and fertile soil [13,14].

The geophilism of *Paracoccidioides* spp. was convincingly established when Albornoz (1971) isolated *P. brasiliensis* from the soil in Venezuela [15]. This isolation was reached few times, demonstrating that the ecological niche continues to be unknown [16,17]. The identification of armadillos, which live in burrows in the soil, infected with *P. brasiliensis* constituted other evidence of the importance of the soil in the life cycle of these fungi [18,19]. However, differently from histoplasmosis [20,21,22] and coccidioidomycosis [23], outbreaks have not been reported in PCM—the first one was recently published, with only eight PCM patients with the AF, after highway construction [24], making the correlation between an exposition and the infection followed by disease difficult.

Barrozo et al. (2009) [25] and Barrozo et al. (2010) [26] have highlighted the importance of environmental variables in influencing the biology of the fungus. These variables influence its survival in the soil, its competitive advantages, its capacity to produce conidia, and the conditions that favor the increase in infection in humans.

PCM incidence is variable among Latin American countries and even within different regions of the same country. However, there is evidence of a relationship between climatic events, such as El Niño, and the spread of several infectious diseases. Nevertheless, it is still challenging to identify PCM’s ecological niche. A connection between the occurrence of the disease and climatic phenomena is not always clear due to the different intensity, temporal and spatial organization, and the fluctuation of climate patterns during different events [27].

A study in the region of Botucatu (São Paulo state, Brazil) has provided strong evidence that the PCM incidence rate is associated with changes in climatic factors, such as absolute air humidity, conditions in the soil, water storage, and the Southern Oscillation Index (SOI). The results have also suggested that fungal growth occurs after an increase in water storage over a long-term period and that there is probably an increase in spore release with increased absolute air humidity in the short term (less than one year) [25].

In order to plan effective control strategies for PCM, it is essential to understand the geoclimatic characteristics of areas at high risk of *Paracoccidioides* infection. We therefore evaluated the distribution of cases of PCM/AF, aiming to evaluate the correlation between this distribution and climatic and environmental factors in the state of MS.

## 2. Materials and Methods

### 2.1. Area, Time Period, and Patients Included in the Study

This study was conducted at the University Hospital of the Federal University of MS, a referral center for PCM treatment. The inclusion criteria encompassed patients diagnosed with the PCM/AF between January 1980 and February 2022.

### 2.2. Origin of the Patients

MS is located in the Central-West region of Brazil and is home to 79 municipalities, primarily located within the Cerrado phytogeographic domain. According to the 2010 demographic census, the population of MS was 2,449,024 inhabitants, with a population density of 6.86 inhabitants per km^2^. The population is composed of 1,091,512 males and 1,067,726 females, with 2,097,238 individuals residing in urban areas and 351,786 residing in rural areas. The state’s economy is predominantly agrarian, focusing on agriculture and livestock activities, notably corn, soy, cotton, and sugar cane cultivation [28,29].

The local government has divided the state into three macroregions, Campo Grande, Dourados, and Três Lagoas, which were further subdivided into 11 microregions: Aquidauana, Campo Grande, Corumbá, Coxim, Jardim, Dourados, Naviraí, Nova Andradina, Ponta Porã, Paranaíba, and Três Lagoas [30].

### 2.3. Characterization of the Patients

The sociodemographic variables selected for the study were sex, address, rural work, age, and date of diagnosis. This information was obtained through a data collection protocol retrospectively applied to medical records from 1980 to 1999 and, subsequently, from 2000 to 2022, the same protocol was prospectively implemented while patients received medical care. 

### 2.4. Analysis of Climatologic Data

Climatological data were obtained from public databases provided by the governments of Brazil (temperature, air humidity, and Niño 3.4 Index) and the United States of America (ocean temperatures in the Niño 3.4 region). The data were then organized in a Excel^®^ Version 2311/2016 (Microsoft Corporation, Redmond, WA, USA) spreadsheet and processed using R Studio software (http://www.rstudio.com/, accessed on 27 December 2023), using the ggplot2 (https://ggplot2.tidyverse.org, accessed on 27 December 2023) and RColorBrewer packages (R-graph-gallery.com, accessed on 27 December 2023). Results are presented as heatmaps, showing each phenomenon’s intensity as a color gradient [31]. PowerPoint^®^ Version 2311/2016 (Microsoft Corporation, Redmond, WA, USA) was used to prepare the figures.

### 2.5. Temperature and Air Humidity Data

The compensated average temperature (°C) and proportion of relative air humidity (RH) were collected from the Meteorology National Institute (Instituto Nacional de Meteorologia—INMET in Portuguese) [32,33]. These values were registered at the meteorological stations of MS from 1981 to 2022. As it was not possible to obtain data for every year between 1981 and 2010, a compilation of nine meteorological stations was used. After 2010, the data became available annually. The compensated average temperature is a mean of the temperatures registered at 9 a.m., 3 p.m., and 9 p.m. on a particular day, along with the maximum and minimum temperatures for that same day. The RH represents the percentage of water vapor in the atmosphere [34].

### 2.6. El Niño Southern Oscillation (ENSO) and Average Global Temperature

The ENSO phenomenon is characterized by index calculations that summarize temperature anomalies at the surface of the sea and other oceanic regions, such as the El Niño 3.4 region in the central portion of the Equatorial Pacific. Anomalies higher than 0.5 °C are denominated La Niña, and those less than −0.5 °C are denominated El Niño. Changes in temperature in the Equatorial Pacific Ocean affect atmospheric circulation patterns, humidity transport, temperature, and precipitation [35]. Data on average ocean temperatures in the Niño 3.4 region were collected from the National Oceanic and Atmospheric Administration (NOAA) website, which compiles and makes climatic information available from oceanic regions around the world [36]. The variation in the Niño 3.4 Index, as well as the occurrence of the El Niño and La Niña phenomena during the study period, was obtained from the Atmospheric Sciences website of the Federal University of Itajubá (Universidade Federal de Itajubá—UNIFEI in Portuguese) [37]. Data from both databases were collected for the period between 1982 and 2020, as the years 1980 and 1981 were not available in the UNIFEI database.

### 2.7. Geospatial Analysis

Patient addresses were converted into geographic coordinates (DD°MM′SS″X/Y) using the Google Earth Pro software (https://earth.google.com, accessed on 27 December 2023) and analyzed using the Quantum GIS (QGIS) 3.14 Pi open-source software. In order to provide a geographical reference for patients with imprecise or absent addresses, the centroid of the given municipality was considered.

Two geospatial analyses were conducted. The first was a thematic map showing density (cases per km^2^), geo-environmental factors, and incidence (cases per 100,000 inhabitants) during the period of the study in the health microregions of MS. For incidence calculations, we used population data from the 2010 national census [38], representing the latest dataset published by the Brazilian Institute of Geography and Statistics (Instituto Brasileiro de Geografia e Estatística—IBGE in Portuguese) within the study period. Shapefiles containing information on Brazilian climate, temperature, and phytogeography were collected from IBGE [39]. These were used to delimit the territorial boundaries of MS for this study. 

The second analysis evaluated PCM/AF cases as a function of time in Campo Grande, the municipality with the highest number of cases. We used satellite images captured by Google Earth Pro. The images were then uploaded into the QGIS software, georeferenced and delimited. We reconstructed the decades evaluated, using thematic maps of the expansion of the urban street network.

We mapped transition zones between different vegetation types in the Campo Grande, which contains most of the recent cases. This information was obtained from the Temporal Vegetation Analysis System (Sistema de Análise Temporal da Vegetação—SATVeg in Portuguese), an online tool developed by the Brazilian Agricultural Research Corporation (Empresa Brasileira de Pesquisas Agropecuárias—EMBRAPA in Portuguese). This tool is intended to access and visualize temporal profiles and vegetational indexes of any region in South America [40].

### 2.8. Statistical Analysis

The chi-square test was used to compare frequencies between two independent groups, and when 20% or more of the expected values were smaller than 5, Fisher’s exact test was used. The chi-square test for one sample was used to analyze the distribution of the patient sex. In order to compare the number of cases per km^2^ (cases/km^2^) and incidence (cases/100,000 inhabitants) among health microregions, we utilized the comparison test for multiple proportions. The distribution of the patients as to admission period and age group was analyzed using Marascuilo’s procedure [41]. Significance was set up at *p* ≤ 0.05.

### 2.9. Ethical Aspects

The project was approved by the Ethics Committee on Human Research of the Federal University of Mato Grosso do Sul under protocol 1620 on 4 March 2010.

## 3. Results

From 1980 to 2022, a total of 747 patients were treated at the Maria Aparecida Pedrossian University Hospital (Campo Grande-MS, Brazil) for PCM. Of these, 81 (10%) presented with the AF. The median occurrence of the AF was 1.9 cases per year, with the highest proportion observed between 1980 and 1989, accounting for 51.9% of all cases. Notably, there was a substantial reduction in the cases of PCM/AF throughout the study period (Table 1). There was a higher proportion of the infection in males when compared to females (67.9%, *p* < 0.001). The male-to-female ratio was 1.5:1.0 for patients up to 13 years old and 4.1:1.0 for those older than 13 years (Table 1). The prevalence of male patients was higher in patients from the oldest group (60.0% to 80.3%; *p* < 0.001). There was no difference in the patient number with regard to the period of admission and age group (Table 1).

Figure 1 shows the distribution of the patients in each health microregion of origin, highlighting Aquidauana, which lies in the west of MS, and Campo Grande. Aquidauana presented the highest incidence rate of cases (7/100,000 inhabitants) and a density of 2.6 cases/km^2^. Campo Grande presented the highest number of cases (34; 42.0%) with an incidence rate of 3.3 cases/100,000 inhabitants and density of 4.4 cases/km^2^. Comparison of health microregions using two references showed that Aquidauana, Coxim, and Jardim occupy an intermediate position in terms of cases/km^2^, while Ponta Porã, Campo Grande, and Dourados do so for cases/100,000 inhabitants (Figure 1).

To investigate the impact of climatic factors, we conducted an analysis to better understand the influence of ENSO events on the frequency of cases from 1980 to 2022. The analysis demonstrated two important El Niño phases in that time as well as a prolonged La Niña phase that lasted from July 1998 to February 2001 (31 months), with a mean temperature of 26.1 °C. Figure 2 shows a noticeable absence of registered cases in the five years after this period. 

The first phase encompassed the years preceding July 1998, and it exhibited the highest number of cases (46; 2.7 cases per year). This coincided with the highest frequency and duration of El Niño phases, which lasted an average of 12.4 months (df = 4.03), with mean temperatures of 27.2 °C. The second phase encompassed the years after February 2001, which showed a significant reduction in the number of cases (19; 0.75 cases per year), with shorter El Niño phases (8 months on average; df = 4.3), and mean temperatures of 27.1 °C.

Figure 3 depicts the climatic data from MS, and it is possible to verify that the highest number of cases was registered during the winter (30–37.0%). In this period, the compensated average temperature was 22.6 °C, with a maximum of 26.9 °C and a minimum of 16.8 °C. During this season, the RH was 60.8%, with a maximum of 76.6% and a minimum of 47.0%. The fall and the winter presented average temperatures of 22.6 °C. However, during the winter, the RH was 11.0% lower than during the fall. In the summer, there were fewer cases (15–18.5%), with an average temperature of 22.6 °C and RH of 73.6%. 

Figure 4 shows the geo-environmental aspects of cases. The phytogeographic domain of Cerrado was the residence origin of 63 (77.8%) cases with PCM/AF. In addition, the soil with loamy texture was the residence origin of 64 (79.0%) cases, and 52 (64.2%) of the cases occurred in areas with temperature ranging from hot to medium (>18 °C in all months in the year). All cases were located in regions with humid climates.

Figure 5 shows the urban boundary of Campo Grande, which is the residence origin of 32.1% of the cases of PCM/AF after 2000, coinciding with the expansion of the urban street network. Analysis conducted with SATVet showed that the phytophysiognomy in this area underwent significant change over three phases. The initial phase was a deforestation process, followed by pasture establishment, and, lastly, a substitution with annual agriculture. 

## 4. Discussion

Although enough knowledge exists regarding PCM to care for patients, very little is known about the ecology of its causative agents [42]. Although isolation of the fungi from its environment has been limited, there are sufficient data available to consider the soil as the natural habitat of the fungi belonging to the genus *Paracoccidioides* and that its primary portal of entry is the respiratory route [7,17,43,44,45,46].

To investigate the environmental aspects associated with PCM, we exclusively analyzed cases of the acute-subacute form, driven by two main reasons, which will be discussed below.

Firstly, because the chronic form has a long latency period, decades may pass between infection and the onset of illness. It was suggested by the diagnosis of patients who lived and worked in rural areas but presented the disease many years after moving to urban areas, working in jobs with no risk for paracoccidioidal infection. Nevertheless, it was confirmed by patients diagnosed many years after migrating from Latin America to countries where PCM does not exist [47]. This long latency period indicates that the residence origin of the patient at diagnosis did not necessarily coincide with the place where the infection was actually acquired. The area where the fungus lives in nature and the patients acquire the infection was named reserve area by Borelli, in order to differentiate from the endemic area, where patients are living at diagnosis [48].

Secondly, because the PCM/AF affects children, adolescents, and young adults, which frequently remain in the same region, the study of the geoclimatic conditions of the areas from where these patients come from can be considered the evaluation of the *Paracoccidioides* reserve area. This was the choice for our study, which was reinforced by the recent publication of a well-documented emergence of PCM/AF after deforestation and massive earth removal during the construction of a highway in Rio de Janeiro (Brazil) [24]. Eight patients from the area of the extensive environmental disturbances were diagnosed, four males and four females, mean age of 22 years old (10–28), with the acute/subacute clinical form and the main clinical findings consisting of lymph node enlargement, hepatomegaly, and splenomegaly. The incidence rate of the PCM/AF in this region was 5.7 times higher than previously observed.

Cases of PCM/AF in MS have decreased over the decades under study. This has already been the subject of publication, where the authors suggest that it is a result of the intense public policy against child labor in the countryside in the 1990s [49]. 

Fungi from the *Paracoccidioides* genus are found in the soil of different regions from Latin America, presenting few natural hosts—human beings and armadillos are considered reservoirs. Although domestic and wild animals have been studied little, infection was serologically confirmed in several of them [50,51,52,53,54,55], but disease was a rare finding [56,57]. As the severity of the clinical picture depends on the virulence of the fungus and the immune response of the patient [8,58], studies on the fungi and the environment where they live should be periodically performed.

Armadillos, burrowing animals that live in forests, fields, and savannas, are naturally infected by several microorganisms pathogenic to humans, such as *Histoplasma capsulatum* var. *capsulatum* [59], *Sporothrix shenckii* [60], *Mycobacterium leprae* [61], *Coccidioidis immitis* [62], and *P. brasiliensis* [18,19], among others. *Dasypus novemcinctus*, the species more frequently infected by *P. brasiliensis*, usually presents no histological findings compatible with tissue lesions. However, two studies reported epithelioid granuloma in the liver, spleen, and lymph nodes [19,60]. 

The relationship between fungi isolated from armadillos and clinical isolates of *P. brasiliensis* was well documented. Homogenates of liver and spleen from armadillos (*Dasypus novemcinctus*) captured in the region of Tucuruí (Pará state, Brazil) were inoculated in hamsters [18] by the intraperitoneal route. These hamsters presented generalized infection by fungi identified as being from the *Paracoccidioides* genus. The inoculated animals presented granulomatous reaction, with macrophages, lymphocytes, epithelioid cells, plasmocytes, and rare giant cells in addition to many parasitic structures. Inoculated mice were sent to our laboratory to perform other evaluations. The isolated fungal cells showed all the characteristics of *P. brasiliensis*, with a strong antigenic power and low virulence to guinea pigs and Wistar rats. The glycoprotein with a molecular weight of 43 kDa, the specific *P. brasiliensis* exoantigen, was easily demonstrated with double immunodiffusion, immunoelectrophoresis, SDS-PAGE, and immunoblotting assays [63]. Finally, the morphology and the pathogenicity to hamsters of this isolate (Pb-T) were compared with those of the Pb-18, a clinical isolate maintained in laboratory. The results demonstrated that Pb-T presented higher values than Pb-18 in the following variables: (a) number of buds by mother cell; (b) sporulation of the mycelial form during eight weeks of culture; (c) mortality rate; (d) antibody serum levels; (e) fungal load; and (f) extent of the lesions in the affected organs. These findings suggested a higher virulence of Pb-T than Pb-18 [64]. The mycological, immunological and pathogenic characteristics, taken together, permit the conclusion that the armadillos (*Dasypus novemcinctus*) constitute a reservoir of the *Paracoccidioides brasiliensis*. These criteria must be taken into consideration in the identification of a *P. brasiliensis* isolate. In a study carried out in the Region of Botucatu, São Paulo state (Brazil), an endemic area of PCM, 5 out of 887 samples isolated from the soil showed similarities to *P. brasiliensis*, including thermo-dependence. However, studies on antigenicity and virulence did not confirm this hypothesis [65]. This finding suggests that armadillos can be a good sentinel animal for the identification of areas in which *P. brasiliensis* lives saprophytically in the soil. 

Studies on intradermal reaction using antigens obtained from strains of the *Paracoccidioides* genus have been carried out to identify endemic areas of PCM [66,67,68,69]. However, the choice of the antigen and its preparation play a crucial role because there are different biochemical compositions, and they will be injected in the population and cross-reactions with similar fungi, such as *Histoplasma capsulatum* var. *capsulatum*, can occur [67]. 

The control of infection in domestic and wild animals, in armadillos as a reservoir, and the studies on specific skin tests demand a great investment and careful design. So, the study of the geoclimatic conditions of different areas, correlating with the human interference in the environment and the incidence of PCM, can greatly contribute to the knowledge of the risk that a case could be PCM, helping physicians to raise this hypothesis in order to reach an early diagnosis and treatment. 

Endemic regions in Colombia are located at different altitudes, between 25 m and 1450 m, with humidity ranging from 65% to 98% and mean temperatures between 21 °C and 29 °C. The incidence of skin-testing reactors in the six regions studied was the same in areas with different geographic and climatic conditions, suggesting that the *P. brasiliensis* development in the soil depends on several other factors, such as the type of the soil and its humidity. Common findings across all areas include patient activities (rural workers who engaged in agriculture or cattle raising) and a predominance of male cases [42].

The influence of climate was also investigated by evaluating the incidence of PCM/AF over the years in the region of Botucatu (São Paulo state, Brazil). We found an increased incidence rate one year after the occurrence of the El Niño phenomena, suggesting that the rise in rainfall leads to a higher humidity of the soil, higher paracoccidioidal sporulation, and, consequently, higher spread of the propagules and infection rates [25,26].

Climatic phenomena such as La Niña and El Niño can affect the rates and distribution of PCM as they cause temperature and rain pattern changes across the planet. For instance, during El Niño years, cloud formation occurs in warmers parts of the ocean. These clouds subsequently move towards the continents, resulting in higher precipitation in South America. This change in rainfall patterns alters the humidity levels of the air and soil [35,70,71].

During prolonged phases of El Niño, we observed a substantial correlation between the increase in the number of PCM cases and higher humidity levels, especially during the wetter seasons. Furthermore, we found a significant predominance of PCM/AF cases in clayey soils and an RH ranging from 60.8 to 73.6%.

On the other hand, during periods of La Niña, the dynamics are reversed, as the decrease in atmospheric and soil humidity makes it impossible for the fungus to multiply and spread. This is the reason underlying the observation of a marked reduction in PCM cases during an extended La Niña period observed in our study (July 1998 to February 2001—31 months).

Another study related the occurrence of PCM to the El Niño phenomenon. Between 2010 and 2012, six new cases of PCM/AF were detected in northeastern Argentina. What is noteworthy is the fact that this area had documented only a single case in the decade preceding these incidents. In addition to the influence of the 2009 El Niño phenomenon, there is a convincing argument that attributes this increase to the construction of a hydroelectric plant and the subsequent creation of its artificial lake. These changes may have altered the environment, increasing air and soil humidity levels in the region [72].

We found a higher number of cases in the phytogeographic region of Cerrado. This may be related to the extensive deforestation which occurred in order to create proper conditions for agriculture and pastures. 

A higher incidence rate of PCM/AF was also observed in the health microregion of Aquidauana, coinciding with the development of agricultural activities in the region. A historic series performed by Ascencio et al. (2020) [73] demonstrated an increase in agriculture, expansion of areas destined for pisciculture, and enlargement of humid areas as well as of urban areas in the region.

Analysis of the urban limits of Campo Grande revealed an emergence of cases of the acute form of PCM in regions where agricultural boundaries were expanding. These findings confirm studies that demonstrate human interference in natural environments and the increased incidence of PCM in highways [24], dams [74], and construction sites, as well as on coffee, corn, and soy plantations [75]. 

Our study has also shown that most cases were diagnosed in the health microregion of Campo Grande, possibly due to the development occurring in that region. Areas which had previously been occupied by forests are now being used by different plantations, which also leads to an increase in the number of rural workers.

The presence of a university hospital, acting as a state reference center for diagnosing and treating PCM, played a crucial role in the development of this study. The hospital’s data revealed the main residence origin of the PCM patients. This allows the health system to support workers; train physicians to identify cases; organize diagnosis protocols, including histopathological examinations, mycological methods and serological tests; provide antifungal compounds for therapy; and follow-up patients, treating any sequelae which may appear. 

This study demonstrates the importance of health services in Brazil, especially in regions undergoing significant human activity, paying special attention to rural workers. PCM is a chronic disease that can lead to several different physical disabilities and sequelae, causing major economic and social impacts in endemic regions. 

One limitation of the study was the difficulty in retrieving daily and hourly climatological data (temperature and RH) for the years prior to 2010. Better and well-maintained databases can lead a better understanding of trends, data, and changes over time. This is useful in the context of health as it helps to understand disease patterns and allows epidemiological predictions to be made and preparations to be put in place.

## 5. Conclusions

This study demonstrated that in MS, patients with the acute/subacute form of paracoccidiodomycosis tended to be residents of the phytogeographic domain of Cerrado, in regions characterized by loamy soils, mainly those destined for cultivation, where native vegetation was supplanted by agricultural activities. Consequently, this change increased human exposure to the fungus.

Understanding the environmental characteristics of the main PCM reserve areas in MS is crucial for anticipating and mapping the healthcare needs for PCM patients. This study contributed to the implementation of the Support Network for the Diagnosis of Fungal Infections (the RADIF acronym in Portuguese). In MS State, RADIF conducts diagnostic consultations and promotes the training of physicians and pathologists in diagnosing fungal infections, including PCM. Special attention is directed towards municipalities in regions considered as reserve areas for the disease

## Figures and Tables

**Figure 1 jof-10-00165-f001:**
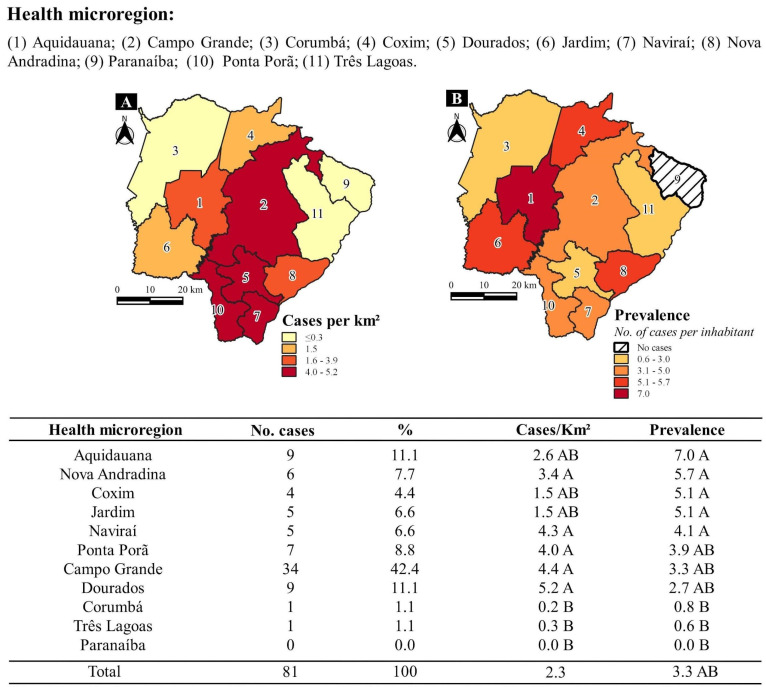
Distribution of 81 patients with the acute/subacute form of paracoccidioidomycosis according to the residence origin at health microregions of Mato Grosso do Sul state. (**A**) Number of cases per square kilometer and (**B**) Prevalence rate: number of patients per 100,000 inhabitants. Capital letters compare values in the same column; values followed by the same letter do not differ from each other (*p* > 0.05), while those followed by different letters show significant differences (*p* ≤ 0.05). Multiple comparison test of proportions.

**Figure 2 jof-10-00165-f002:**
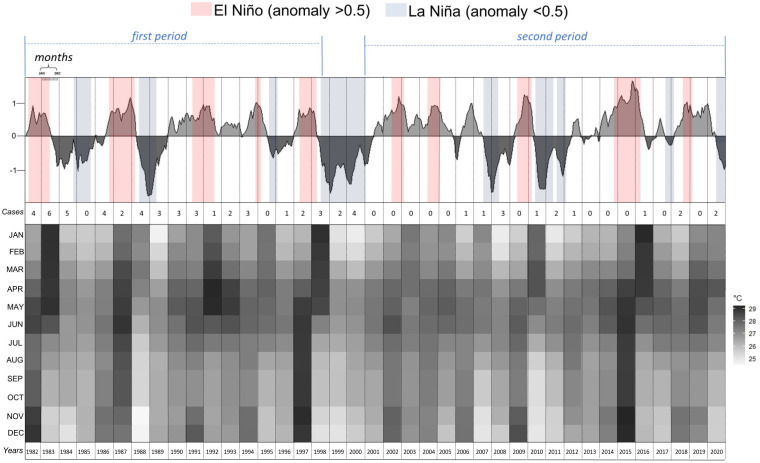
Cases of the acute/subacute form of PCM from Mato Grosso do Sul state according to the climatic anomalies (ENSO—3.4) from 1982 to 2020.

**Figure 3 jof-10-00165-f003:**
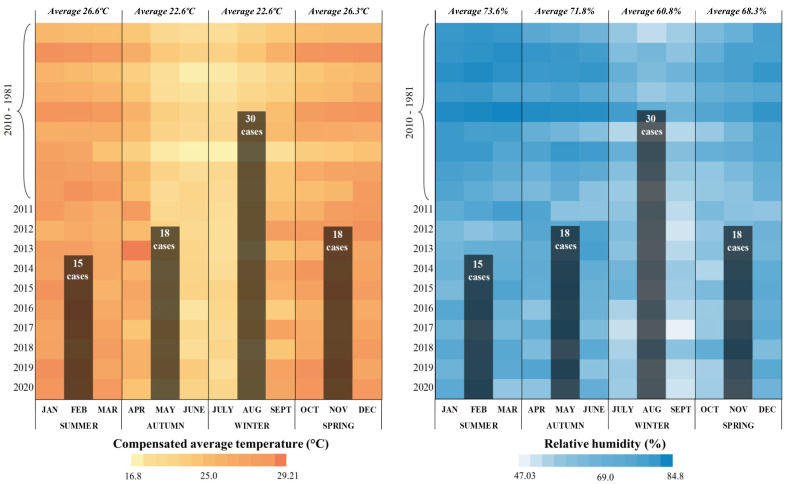
Cases of the acute/subacute form of paracoccidioidomycosis compared with compensated average temperature (°C) and relative air humidity (RH) registered between 1980 and 2020 in Mato Grosso do Sul state (Brazil).

**Figure 4 jof-10-00165-f004:**
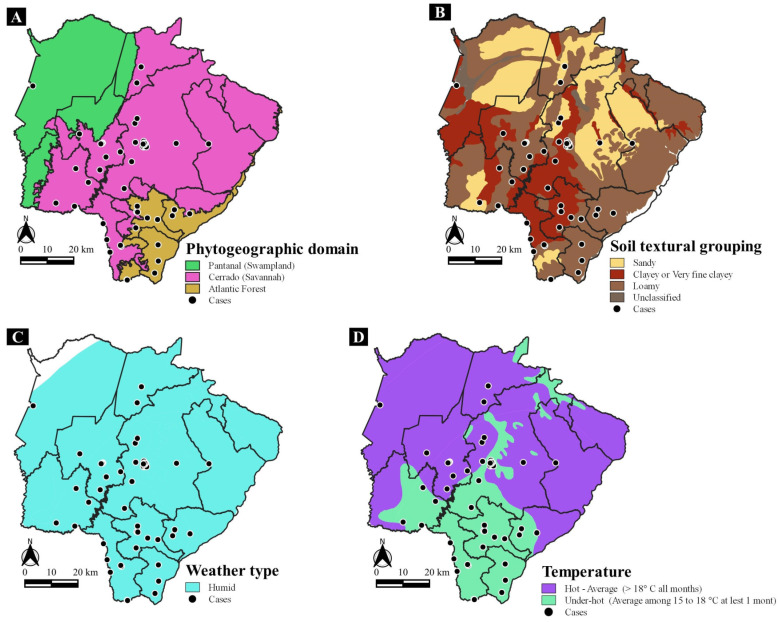
Spatial distribution of the acute/subacute cases of paracoccidioidomycosis according to (**A**) Phytogeographic domain, (**B**) Soil textural grouping, (**C**) Weather type and (**D**) Temperature aspects of the health microregions from Mato Grosso do Sul (Brazil).

**Figure 5 jof-10-00165-f005:**
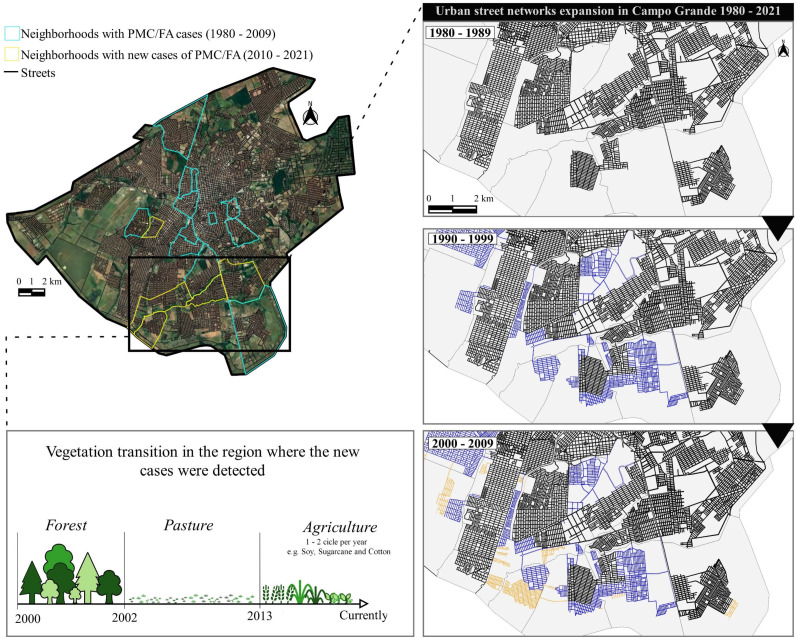
Case distribution of the acute/subacute form of paracoccidioidomycosis in the urban perimeter in the municipality of Campo Grande (Mato Grosso do Sul state, Brazil).

**Table 1 jof-10-00165-t001:** Distribution of 81 cases of the acute/subacute form of PCM according to sex, age group, year of admission, and rural activity.

Variable	Total (n = 81)	Age Group ≤13 Years Old (n = 15)	Age Group >13 Years Old (n = 66)	*p*-Value
Sex				
Male	62 (76.5)			<0.01 *
Female	19 (23.5)			
Male (M)		09 (60.0)	53 (80.3)	<0.001 **
Female (F)		06 (40)	13 (19.7)	
M: F rate	3.3:1.0	1.5:1.0	4.1:1.0	
Admission period				
1980–1989	42 (51.9)	08 (53.3)	34 (51.5)	>0.05 ***
1990–1999	21 (25.9)	02 (13.3)	19 (28.8)	
2000–2009	08 (09.9)	02 (13.3)	06 (09.1)	
2010–2019	06 (07.4)	01 (06.8)	05 (07.6)	
2020–2021 #	04 (04.9)	02 (13.3)	02 (03.0)	
Rural activity	35 (43.2)	03 (20.0)	32 (48.5)	<0.0001 ****

# This period contains only two years and therefore was excluded from statistical analysis. ( )—percentage, n—number of patients. * Prevalence as to sex (x^2^ test for one sample). ** Comparison of prevalence as to sex and age group (Fisher’s exact test). *** Comparison of prevalence as to age group and admission period (Marascuilo’s procedure). **** Comparison of the prevalence of rural activities as to age groups.

## Data Availability

Data are contained within the article.

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
