# Peer review of "The Effect of Geoclimatic Factors on the Distribution of Paracoccidioidomycosis in Mato Grosso do Sul, Brazil"

_jof, 2024, doi:10.3390/jof10030165_

Round 1

Reviewer 1 Report

Comments and Suggestions for Authors

Table 1 is not understood. If the total number of male patients is 55, why adding the minors and those over 13 years of age gives 62??

Likewise, the values of the female sex do not coincide. If you say that the total number of female patients is 26, because the sum below gives 19??

Discussion

It is a good study poorly discussed. The discussion must be expanded. You have obtained many results that have not been discussed.

2nd paragraph in unclear.

Line 266  in these six regions… which six regions?????

If the incidence of skin-testing reactors WAS THE SAME in areas with different geographic and climatic conditions…,  why do you suggest that the P. brasiliensis development in the soil depends on several factors…

Distribution of your 81 cases over 42 years must be discussed. The incidence of PCM/AF over the years must be discussed.

Outbreaks of PCM as a consequence of the El Niño phenomena and climatic changes have been detected and already reported, and not only in Brazil. This should be discussed in this manuscript with the following reports:

·         Barrozo LV, Mendes RP, Marques SA et al. Climate and acute/subacute paracoccidioidomycosis in a hyper-endemic area in Brazil. 2009;38(6):1642–9.

·         Barrozo LV, Benard G, Siqueira Silva ME et al. First description of a cluster of acute/subacute paracoccidioidomycosis cases and its association with a climatic anomaly. PLoS Negl Trop Dis 2010; 4(3): e643. doi: 10.1371/journal.pntd.0000643.

·         Giusiano, G.; Aguirre, C.; Vratnica, C.; Rojas, F.; Corallo, T.; Cattana, M.E.; Fernández, M.; Mussin, J.; Sosa, M.A. Emergence of acute/subacute infant-juvenile paracoccidioidomycosis in Northeast Argentina: Effect of climatic and anthropogenic changes? Med. Mycol. 2019, 57, 30–37.

The effect of El Niño compared to the effect of La Niña on PCM needs to be better explained and related/discussed with the your findings.

Conclusion

Specifically. What is your conclusion about the correlation between PCM distribution and climatic and environmental factors? You only conclude about deforestation

What would be the importance of obtaining daily and hourly climatological data (temperature and RH)?  during the 42 years? How would you evaluate/discuss daily and/or hourly?

Line 74. You say …In order to plan effective control strategies for PCM, it is essential to understand the geoclimatic characteristics of areas at high risk of Paracoccidioides infection. So… considering your study and results, which control measure(s) would you plan?

Comments on the Quality of English Language

Minor editing of English language required

Author Response

Thank you for your comments. Below are our answers, which have also been inserted into the body of the manuscript point-by-point.

Likewise, the values of the female sex do not coincide. If you say that the total number of female patients is 26, because the sum below gives 19??

Yes, thank you for your observation. I have addressed the highlighted errors and made the necessary corrections.

2nd paragraph in unclear.

Thank you for your comment. We have now adjusted it:

LINE 358

               (…) To investigate the environmental aspects associated with PCM, we exclusively analyzed cases of the acute-subacute form, driven by two main reasons, which will be discussed below.

               Firstly, because the chronic form has a long latency period, decades may pass between infection and the onset of illness. (…)

Line 266 …in these six regions… which six regions?????

If the incidence of skin-testing reactors WAS THE SAME in areas with different geographic and climatic conditions…, why do you suggest that the P. brasiliensis development in the soil depends on several factors…

The six regions are the ones investigated in the study conducted in Colombia. However, you are correct; the sentence was not clear. We made a small adjustment.

The findings suggest that other unexplored factors could play a role, such as soil type and moisture. We completed the sentence.

LINE 463

(…) The incidence of skin-testing reactors in the six regions studied was the same in areas with different geographic and climatic conditions, suggesting that the P. brasiliensis development in the soil depends on other several factors, such as the type of the soil and its humidity.(…)

Distribution of your 81 cases over 42 years must be discussed. The incidence of PCM/AF over the years must be discussed.

Yes, we agree. Our group has published a paper on this subject. We have now inserted a paragraph citing it.

LINE 388

Cases of PCM/AF in MS have decreased over the decades under study. This has already been the subject of publication, where the authors suggest that it is a result of the intense public policy against child labor in the countryside in the 1990s.

Outbreaks of PCM as a consequence of the El Niño phenomena and climatic changes have been detected and already reported, and not only in Brazil. This should be discussed in this manuscript with the following reports:

Excellent suggestion. We had already discussed the first two articles, so in this version, we added the third suggested article (Giusiano et al., 2019), which significantly enriched our discussion.

LINE 425

               Another study related the occurrence of PCM to the El Niño phenomenon. Between 2010 and 2012, six new cases of PCM/AF were detected in northeastern Argentina. What is noteworthy is the fact that this area had documented only a single case in the decade preceding these incidents. In addition to the influence of the 2009 El Niño phenomenon, there is a convincing argument that attributes this increase to the construction of a hydroelectric plant and the subsequent creation of its artificial lake. These changes may have altered the environment, increasing air and soil humidity levels in the region (Gisiano et al, 2019).

The effect of El Niño compared to the effect of La Niña on PCM needs to be better explained and related/discussed with the your findings

LINE 484

(..) During prolonged phases of El Niño, we observed a substantial correlation between the increase in the number of PCM cases and higher humidity levels, especially during the wetter seasons. Furthermore, we found a significant predominance of PCM/AF cases in clayey soils, with a RH ranging from 60.8 to 73.6%.

               On the other hand, during periods of La Niña, the dynamics are reversed, as the decrease in atmospheric and soil humidity makes it impossible for the fungus to multiply and spread. This is the reason underlying the observation of a marked reduction in PCM cases during an extended La Niña period observed in our study (July 1998 to February 2001 - 31 months).

 Specifically. What is your conclusion about the correlation between PCM distribution and climatic and environmental factors? You only conclude about deforestation

(…) Thank you for your comment. We adjusted the paragraph to make it clearer that climatic and environmental factors can favor soil conditions for planting, and agricultural activity exposes humans to the fungus.

LINE 549

               This study demonstrated that in Mato Grosso do Sul (MS), patients with the acute/subacute form of paracoccidiodomycosis tended to be residents of the phytogeographic domain of Cerrado, in regions characterized by loamy soils, mainly those destined for cultivation, where native vegetation was supplanted by agricultural activities. Consequently, this change increased human exposure to the fungus.

               Understanding the environmental characteristics of the main PCM reservareas in MS is crucial for anticipating and mapping the healthcare needs for PCM patients. This study contributed to the implementation of the Support Network for the Diagnosis of Fungal Infections (RADIF acronym in Portuguese) in Mato Grosso do Sul State, RADIF conducts diagnostic consultations and promotes the training of physicians and pathologists in diagnosing fungal infections, including PCM. Special attention is directed towards municipalities in regions considered as reservarea for the disease.

 What would be the importance of obtaining daily and hourly climatological data (temperature and RH)? during the 42 years? How would you evaluate/discuss daily and/or hourly?

We had previously highlighted the significance of understanding the regions where PCM patients reside for the purpose of organizing the healthcare system (original version). In the current version, we added to the conclusions that the study facilitated the recent establishment of RADIF, a support network for diagnosing fungal infections in the state. RADIF conducts diagnostic consultations and promotes the training of physicians and pathologists in diagnosing fungal infections, including PCM. Special attention is directed towards municipalities in regions considered reservarea for the disease.

Reviewer 2 Report

Comments and Suggestions for Authors The paper is original, however, I think that the authors could enrich
the discussion, with the epidemiology of
Paracoccidioidomycosis
in neighboring countries, Venezuela, which has a significant
rate of clinical cases, and Colombia.

Author Response

The paper is original, however, I think that the authors could enrich the discussion, with the epidemiology of Paracoccidioidomycosis in neighboring countries, Venezuela, which has a significant rate of clinical cases, and Colombia.

Thank you for your comment. Our primary objective in this article is to concentrate on the geoclimatic aspects of PCM. Therefore, we have selected articles specifically addressing this focus for discussion. References 15 and 38  pertain to Venezuela and Colombia, respectively. Additionally, in line with your suggestion, this revised version incorporates a study from Argentina that was not previously addressed in the original version (Giusiano et al, 2019).

Reviewer 3 Report

Comments and Suggestions for Authors

The manuscript presents a very well designed and conducted study on the distribution of PMC in Mato Grosso, Brazil. Big strengths are the large study period of time and the availability of geoclimatic data for almost the entire study period. 

I just have a few minor questions:

Population structure may have varied along the study period and this will alter incidence calculations. Why did the authors take 2010 as a representative year?

Was it done for the entire 42 years period? “This information was obtained from a service protocol that was completed while the patient was receiving outpatient care”

What did you use climate data from USA for?

Temperature and air humidity data: What about 1980?

El Niño Southern Oscillation (ENSO) and average global temperature: What about 1980 and 1981?

“…very little is known about the ecology of its causative agents”. Reference 21 dates from 1968. There are newer publications trying to associate geoclimatic factors with PMC ecology. It would be advisable to update the bibliography for this sentence.

Author Response

Dear reviewer, thank you for your comments. Below are our responses, inserted in the body of the manuscript, point-by-point.

Population structure may have varied along the study period and this will alter incidence calculations. Why did the authors take 2010 as a representative year?

Line 208

(…) For incidence calculations, we used population data from the 2010 national census, representing the latest dataset published by the Brazilian Institute of Geography and Statistics (Instituto Brasileiro de Geografia e Estatística – IBGE in Portuguese) within the study period (…)

 Was it done for the entire 42 years period? “This information was obtained from a service protocol that was completed while the patient was receiving outpatient care”

Line 153

(…) This information was obtained through a data collection protocol retrospectively applied to medical records from 1980 to 1999 and, subsequently, prospectively, from 2000 to 2022, the same protocol was prospectively implemented while patients received medical care.

What did you use climate data from USA for?

LINE 158

Climatological data were obtained from public databases provided by the governments of Brazil (temperature, air humidity, and Niño 3.4 Index) and the United States of America (ocean temperatures in the Niño 3.4 region).

LINE 189

(…) Data on average ocean temperatures in the ENSO 3.4 region were collected on the National Oceanic and Atmospheric Administration (NOAA) website, which compiles and makes available data with climatic information from oceanic regions around the world. The variation of the Niño 3.4 Index as well as the occurrence of the El Niño and La Niña phenomena during the study period were collected on the Atmospheric Sciences website of the Federal University of Itajubá (UNIFEI).

Temperature and air humidity data: What about 1980?

LINE 173

(…) Data from 1980 were not available (…)

El Niño Southern Oscillation (ENSO) and average global temperature: What about 1980 and 1981?

LINE 196

Data from both databases were collected for the period between 1982 and 2020, as the years 1980 and 1981 were not available in the UNIFEI database.

 “…very little is known about the ecology of its causative agents”. Reference 21 dates from 1968. There are newer publications trying to associate geoclimatic factors with PMC ecology. It would be advisable to update the bibliography for this sentence.

(…) In the current version we have added two more recent references.

Round 2

Reviewer 1 Report

Comments and Suggestions for Authors

I consider that the observations and suggestions were revised

Author Response

We would like to thank you for the comments and suggestions.